# Classification of Lifting Techniques for Application of A Robotic Hip Exoskeleton

**DOI:** 10.3390/s19040963

**Published:** 2019-02-25

**Authors:** Baojun Chen, Francesco Lanotte, Lorenzo Grazi, Nicola Vitiello, Simona Crea

**Affiliations:** 1The BioRobotics Institute, Scuola Superiore Sant’Anna, 56127 Pisa, Italy; francesco.lanotte@santannapisa.it (F.L.); lorenzo.grazi@santannapisa.it (L.G.); nicola.vitiello@santannapisa.it (N.V.); simona.crea@santannapisa.it (S.C.); 2Fondazione Don Carlo Gnocchi, 20148 Milan, Italy

**Keywords:** lifting detection, pattern recognition, hip exoskeleton, exoskeleton control

## Abstract

The number of exoskeletons providing load-lifting assistance has significantly increased over the last decade. In this field, to take full advantage of active exoskeletons and provide appropriate assistance to users, it is essential to develop control systems that are able to reliably recognize and classify the users’ movement when performing various lifting tasks. To this end, the movement-decoding algorithm should work robustly with different users and recognize different lifting techniques. Currently, there are no studies presenting methods to classify different lifting techniques in real time for applications with lumbar exoskeletons. We designed a real-time two-step algorithm for a portable hip exoskeleton that can detect the onset of the lifting movement and classify the technique used to accomplish the lift, using only the exoskeleton-embedded sensors. To evaluate the performance of the proposed algorithm, 15 healthy male subjects participated in two experimental sessions in which they were asked to perform lifting tasks using four different techniques (namely, squat lifting, stoop lifting, left-asymmetric lifting, and right-asymmetric lifting) while wearing an active hip exoskeleton. Five classes (the four lifting techniques plus the class “no lift”) were defined for the classification model, which is based on a set of rules (first step) and a pattern recognition algorithm (second step). Leave-one-subject-out cross-validation showed a recognition accuracy of 99.34 ± 0.85%, and the onset of the lift movement was detected within the first 121 to 166 ms of movement.

## 1. Introduction

Heavy load lifting is a common task in warehouses and manufacturing environments. The sixth European Working Conditions Survey carried out in 2015 in 35 countries revealed that 32% of workers perform tasks like carrying or moving heavy loads, and 10% of health and personal care workers (e.g., nurses) are required to lift and carry patients [1]. In particular, repetitive lifting of heavy loads exposes workers to the risk of physical ailments such as low back pain [2] and other work-related musculoskeletal disorders [3]. To reduce the risk of injuries caused by heavy load lifting, numerous passive [4,5,6,7] and active [8,9,10,11,12,13,14,15,16,17] exoskeletons have been developed by several research groups and companies worldwide over the last decade.

Passive exoskeletons rely on spring-based mechanisms and elastic materials and are usually designed to store energy during the bending phase of human movement and release it during the lifting phase. Typically, passive systems are lightweight and, in many cases, have been proven effective in reducing the muscular activity of the lower back when performing lifting tasks [4,5,6,18,19,20,21]. As a drawback, they have limited versatility and can be perceived as uncomfortable in tasks that were not intended to be assisted, such as walking [19,22,23,24]. By contrast, active lumbar devices can produce net positive work to extend the user’s trunk, but at the price of heavier weight and encumbrance. The assistive action can be accomplished by designing adaptive force or torque profiles that vary based on the type and speed of the movement performed by the user. To fully exploit the potential advantages of active exoskeletons, control systems must be able to detect the onset of the lifting movement and generate task-appropriate assistance profiles on a dynamic, as-needed basis. To fulfill this aim, in our previous studies, we developed a real-time rule-based algorithm to detect the onset of the lifting movement, and we successfully validated the algorithm with an active hip exoskeleton [25,26].

To further advance the state of the art among lift detection algorithms, real-time classification of different lifting techniques is essential to providing the most appropriate assistive action. The lifting technique is defined by the posture adopted just before the load is lifted [27,28,29], and the selection of lifting technique by workers is typically made based on several factors, such as the initial position [30] and weight [31] of the load.

Among symmetric lifting techniques (i.e., those not involving twisting of the trunk), the most common are so-called squat and stoop lifting. In squat lifting, the knees are fully flexed while the trunk is held as straight as possible to reach the load [27]. In stoop lifting, the knees are almost fully extended and the trunk is bent forward. Squat lifting is the most clinically recommended lifting technique, as the compression force at the lumbosacral (L5/S1) disks has a relatively small peak value, but it is not well-suited to prolonged and repetitive lifting, because it concentrates loads on the knee muscles, which are not as strong as the hip and trunk muscles [32]. On the other hand, stoop lifting produces a relatively high peak compression force on the L5/S1 joint [33]. Asymmetric lifting, achieved by twisting the trunk, is usually preferred by workers when the load is not placed in front of them [34], but it is not recommended by ergonomists due to the high moment exerted at the L5/S1 junction by the erector spinae muscles [35].

In this paper, we present a two-step algorithm to detect the onset of lifting movements and classify the adopted technique among squat lifting, stoop lifting, left-asymmetric lifting, and right-asymmetric lifting. To the best of our knowledge, there are no previous studies attempting to recognize different lifting techniques for the control of lumbar exoskeletons. To evaluate the performance of the proposed algorithm, fifteen subjects were recruited and asked to perform lifting tasks while wearing an active pelvis orthosis (APO). Two hip joint encoders and an inertial measurement unit (IMU) on the backpack of the APO were used for onset detection and technique recognition. A subject-independent offline analysis was conducted to investigate the generalization capabilities of the proposed algorithm.

## 2. Materials and Methods

### 2.1. Experimental Setup

The APO (Figure 1A) is a lightweight hip joint exoskeleton designed for assisting hip flexion and extension movements; it is an improved version of the system presented in [36] and has been recently used in other studies for lift assistance [25,26,37]. The APO is composed of (i) a frame structure connected to the user’s trunk by means of a lumbar orthopedic cuff, straps, and braces, and (ii) two rotating links connected to the user’s thighs. The two links are actuated by two series elastic actuators, which can provide up to ±22 N·m of peak torque. Two absolute encoders measure the hip joint angles. The APO’s control electronics are housed in a “backpack” module integrated into the lumbar orthosis.

A 9 degree-of-freedom (DOF) IMU (LSM9DS0, STMicroelectronics Inc., Geneva, Switzerland) was placed on the backpack of the exoskeleton (Figure 1A). A sensory fusion algorithm was used to calculate Euler angles using raw IMU signals in real time [38], with the roll angle and yaw angle describing trunk motion in the sagittal and transverse planes, respectively. All sensors were sampled at 100 Hz. A user datagram protocol (UDP) link was set to synchronize the data collected from the APO and the IMU, which were both connected by Ethernet cables to a switcher, connected in turn to the main control laptop running the graphical user interface.

### 2.2. Experimental Protocol

The study was carried on at the premises of The BioRobotics Institute of Scuola Superiore Sant’Anna (Pontedera, Italy), approved by the local Ethics Committee (approval n. 1/2018), and conducted in accordance with the principles stated in the Declaration of Helsinki. Fifteen healthy male subjects were recruited and provided written informed consent to participate in the study (age: 27.4 ± 2.4 years, height: 177.6 ± 8.1 cm, weight: 69.7 ± 7.4 kg).

During the experimental session, the APO was used in transparent mode (i.e., closed-loop torque control with a reference torque of zero [25]). Once wearing the device, subjects were asked to stand still for three seconds to calibrate the estimated trunk Euler angles from the IMU.

The experiment consisted of two sessions and used the experimental setup shown in Figure 1B. In Session 1, subjects were asked to perform repetitive lifting according to four different techniques, namely, squat lifting, stoop lifting, left-asymmetric lifting, and right-asymmetric lifting. For each technique, subjects lifted up and set down a 5-kg box for 10 repetitions, upon visual cues from a graphical interface in front of the participant. Subjects were instructed to lower their bodies in the technique-appropriate manner to reach the load on a 20-cm-high stand, lift the load, put it on a 75-cm-high table, return to a standing posture, and wait for the next visual cue. After a new visual cue was displayed, they were then instructed to reach the load on the table, lower it down, place it on the stand, and return back to standing. Subjects performed the lifting and lowering movements at their self-selected speeds.

In Session 2, subjects were asked to perform the following movement sequence: starting from a seated position on a 45-cm-high stool, they were requested to stand up, walk to the load, lift it up, place it on a table, return to standing posture, reach the load again, lower it down, walk back to the stool, and sit down. All load lifting and lowering movements were triggered by visual cues, and this full sequence was repeated 4 times for each lifting technique.

### 2.3. Recognition Algorithm

A two-step recognition strategy was used to detect the onset of the lifting movement and recognize the adopted lifting technique (Figure 2). The first step of the algorithm was designed to detect the lifting movement using the hip joint angle measurements, according to the algorithm described in [25]. Then, the second step aimed to classify the lifting technique or recognize false positive lifting movements identified by the first step. The classification was performed using a quadratic discriminant analysis (QDA) classifier based on signals collected by hip joint encoders and the IMU on the APO backpack. A more detailed description of the two-step algorithm is provided in the following sections.

#### 2.3.1. First-Step Algorithm

The first-step algorithm uses a rule-based strategy to distinguish two primary indicator movements from all the movements the user can do―namely, (i) the *Pre-extension* movement, i.e., the quasi-static (trunk or hip) flexion phase, which occurs prior to the lift phase, typically when the person grasps or releases the object, and (ii) the *Extension* movement, i.e., the lift phase, which can involve mostly the trunk or the hip extension, depending on the lifting technique. All movements not classified as *Pre-extension* or *Extension* are classified as *Other* [25].

At each time iteration i, if the current classification state is *Other*, the transition to *Pre-extension* is governed by Rule 1, which requires all three of the following conditions to be satisfied:(1)$$\theta_{diff}\left(i\right)<h_{1},$$
(2)$$\theta_{mean}\left(i\right)>h_{2},$$
(3)$$\bar{\theta_{std}}\left(i\right)=std\left(\theta_{mean}\left(\left[i-9:i\right]\right)\right)<h_{3}.$$

$\theta_{diff}\left(i\right)$ is defined as $\left|\theta_{L}\left(i\right)-\theta_{R}\left(i\right)\right|$, where $\theta_{L}\left(i\right)$ and $\theta_{R}\left(i\right)$ denote the left and right hip joint angles; $\theta_{mean}\left(i\right)$ is defined as $(\theta_{L}\left(i\right)+\theta_{R}\left(i\right))/2$ and $\bar{\theta_{std}}\left(i\right)$ is the standard deviation of $\theta_{mean}$ over a moving window of 10 samples (i.e., 100 ms); $h_{1}$, $h_{2}$ and $h_{3}$ are empirically predefined thresholds.

When the current state is *Pre-extension*, Rule 2 is used to detect the transition to *Extension*, based on the following conditions (assuming $\;t_{0}$ as the time when the transition from *Other* to *Pre-extension* occurred):(4)$$\bar{\theta_{std}}\left(i\right)>h_{4},\;\mathrm{AND}$$
(5)HasPeak(i)=1, AND
(6)$$T_{PE}\left(i\right)\leq T_{extension}.$$

$T_{PE}\left(i\right)$ is the current duration of the *Pre-extension* phase, i.e., the time passed since $\;t_{0}$; HasPeak(i) equals 1 if a local maximum of $\theta_{mean}$ has been detected since $t_{0}$—otherwise it equals 0; $h_{4}$ and $T_{extension}$ are predefined thresholds.

Rule 3 allows the algorithm to transition from *Pre-extension* to *Other*, if the *Pre-extension* duration is longer than a certain threshold $T_{extension}$:(7)$$T_{PE}\left(i\right)>T_{extension}.$$

Rule 4 determines the transition from *Extension* to *Other*, whenever two conditions are met:(8)$$\theta_{mean}\left(i\right)<h_{5},\;\mathrm{AND}$$
(9)$$\bar{\theta_{std}}\left(i\right)<h_{6}\;\mathrm{or}\;HasValley\left(i\right)=1.$$

HasValley(i) equals 1 if a local minimum of $\theta_{mean}$ has been detected since $t_{1}$ (defined as the time when the transition from *Pre-extension* to *Extension* occurred)—otherwise it equals 0; $h_{5}$ and $h_{6}$ are predefined thresholds.

#### 2.3.2. Second-Step Algorithm

According to our previous study [25], the rule-based algorithm is able to avoid false positive lift recognitions in most static activities (such as still sitting or quiet standing) and dynamic activities such as walking or turning around. Most recognition errors occur during the stand-to-sit transition—in particular, after the touch-down event, when trunk extension occurs to complete the transition, the rule-based algorithm classifies the movement as *Extension*. Such false positive detection can lead to an improper and uncomfortable assistive torque delivered by the robot.

To improve the algorithm robustness and avoid false recognitions (in particular, those occurring in stand-to-sit movements), a second-step algorithm based on a QDA classifier has been implemented. It acts as soon as the first-step outputs *Extension* and aims to recognize five lifting classes, namely, no-lift (in case of false positive *Extension* detections), and the four lift techniques, i.e., squat, stoop, left-asymmetric, and right-asymmetric lifting. For all the non-lift movements investigated in this study, *Extension* was only detected during stand-to-sit movements. Therefore, the no-lift classification was achieved only in stand-to-sit movements.

To extract relevant information from the sensor signals, seven candidate features for the QDA classifier were computed (Table 1) to generate the initial feature pool. According to the kinematic signals collected by the on-board sensors, these features have relatively large variance among the different techniques and could provide useful information for lifting technique classification (Figure 3). For instance, visual inspection of signals revealed that in squat and stoop lifting, the left and right hip angles have similar profiles, whereas they differ significantly for the asymmetric lifts. Similarly, squat and stoop lifting are differentiated by the roll angle (the angle between the trunk and vertically upward direction) profile, which is similar to the hip joint angles (180°—the angle between the trunk and the thigh) for stoop lifting yet reaches significantly lower peak values during squat lifting. Moreover, it is worth noting that $\theta_{thigh}$ could be considered as a rough estimate of the thigh angle (the angle between the thigh limb and vertically downward direction) in the sagittal plane, which was considered useful to discriminate sitting down from lifting.

Meanwhile, left-asymmetric and right-asymmetric lifting are characterized by a trunk yaw angle variance which is higher than symmetric lifting, and the trunk yaw angle reaches different values when trunk is twisted towards left or right. In order to compensate for gross variations in the yaw angle (determined by magnetometer) due to changes in the subject’s spatial orientation, we adjusted it by subtracting the mean value of yaw angle over the last 500 ms before the *Pre-extension* phase (i.e., 50 samples before $t_{0}$), which could be regarded as an offset related with subject’s orientation.

A greedy-forward selection algorithm, similar to the one in [39], was used for feature set determination, performed as follows: each single feature was used for recognition, and the one with the highest accuracy was selected as the first feature of the feature set. Then each feature in the unselected feature pool was combined with the selected feature for recognition, and the hybrid feature with the best performance was selected as the second feature of the feature set. The process was repeated until the optimal feature set was determined. The optimal feature set was used by the classifier in the second step for lifting technique classification. To avoid overfitting, the feature selection was stopped if the added feature leads to an increment of overall recognition accuracy lower than 0.1%. Note that the optimal feature set was selected based on the average accuracy over all subjects.

### 2.4. Performance Evaluation

For the purpose of controlling exoskeletons based on the recognized movement, the speed (delay) of movement detection and accuracy of technique recognition are crucial metrics to evaluate the performance of the proposed algorithm. In this evaluation, the data collected in Session 1 and Session 2 are combined to generate the dataset for each subject.

#### 2.4.1. Lifting Technique Recognition

Recognition accuracy (RA) describes the overall reliability of the algorithm for lifting technique recognition. It was calculated as:(10)$$\mathrm{RA}=\;\frac{N_{correct}}{N_{total}},$$
where $N_{correct}$ is the number of correctly recognized movements and $N_{total}$ is the total number of test movements. A confusion matrix (CM) was then calculated to quantify the error distribution across classes:(11)$$\mathrm{CM}=\;\left[\begin{array}{ccc} c_{11} & c_{12} & \begin{array}{cc} \ldots & c_{15} \end{array}\\ c_{21} & c_{22} & \begin{array}{cc} \ldots & c_{25} \end{array}\\ \begin{array}{c} \vdots \\ c_{51} \end{array} & \begin{array}{c} \vdots \\ c_{52} \end{array} & \begin{array}{cc} \ddots & \vdots \\ \ldots & c_{55} \end{array} \end{array}\right].$$

Each element of the matrix is defined as:(12)$$c_{ij}=\frac{n_{ij}}{{\bar{n}}_{i}},$$
where $n_{ij}$ is the number of movements in class i that recognized as class j. ${\bar{n}}_{i}$ is the total number of movements in class i. Therefore, off-diagonal numbers indicate misrecognition rates.

Apart from recognition accuracy, sensitivity and specificity were also calculated as follows:(13)$$\mathrm{Sensitivity}\left(k\right)=\;\frac{TP_{k}}{TP_{k}+FN_{k}},$$
(14)$$\mathrm{Specificity}\left(k\right)=\;\frac{TN_{k}}{TN_{k}+FP_{k}},$$
where $TP_{k}$, $FP_{k}$, $TN_{k}$, and $FN_{k}$ are the numbers of true-positive, false-positive, true-negative, and false-negative for class k, respectively.

To evaluate the generalization capabilities of the proposed algorithm, the four metrics mentioned above were calculated for subject-dependent and subject-independent recognition. For subject-dependent recognition, 5-fold cross-validation (CV) was performed for each subject. Data of each subject were randomly divided into 5 folds, with each fold having similar numbers of movements in each class. One fold was selected as the testing dataset and the remaining 4 folds as the training dataset. This procedure was repeated 5 times and each fold of the data was used as the testing dataset only once. Recognition metrics for each subject were the average values of corresponding metrics obtained from the 5 repetitions. The overall performance of subject-dependent recognition was calculated by averaging the individual performances of all the subjects.

For subject-independent recognition, leave-one-subject-out cross-validation was carried out to evaluate the algorithm performance. Data of one subject served as the testing dataset, and data of the remaining 14 subjects as the training dataset. The classification metrics were calculated 15 times, each one using one subject’s data as the test dataset. The overall performance of subject-independent recognition was the average value of performances obtained from the 15 repetitions. A paired-samples t-test analysis with a confidence interval of 95% was performed to compare the performance of subject-dependent and subject-independent recognition.

#### 2.4.2. Lifting Detection Delay

We assumed that subjects started the lifting movement when a local maximum of hip flexion angle was observed in *Pre-extension* phase, and we thus defined the absolute time delay for lifting detection as:(15)$$T=t_{1}-t_{peak},$$
where $t_{peak}$ is the moment of hip angle peak in the *Pre-extension* phase, i.e., the reference event.

In the experiment, subjects performed the lifting movement at their preferred speed. However, in our previous study [25], we noticed detection delay was significantly influenced by lifting speed. To avoid the impact of lifting speed on recognition time evaluation, normalized detection delay was calculated as:(16)$$T_{n}=\frac{T}{t_{2}-t_{peak}},$$
where $t_{2}$ is the terminal moment of the *Extension* phase detected by the algorithm in the transition from the *Extension* to the *Other* phases. $T_{n}$ could thus be considered as the percentage of lifting movement duration.

Lifting detection delay was evaluated with the result of subject-independent recognition. The set of average detection delays for each subject was calculated separately for each lifting technique. One-way repeated measures ANOVA was then performed to investigate whether lifting technique had a significant impact on lifting detection delay. Next, paired samples t-tests were run to detect significant differences in detection delays between specific pairs of lifting techniques. For all comparisons, the confidence interval percentage was set to 95%.

## 3. Results

### 3.1. Lifting Technique Recognition

For subject-dependent recognition, the optimal feature set identified was $\left\{\alpha_{thigh},\delta_{LR}\;,\delta_{thigh}\;,\;\psi_{adj}\right\}$. The overall recognition accuracy across 15 subjects was 99.65 ± 0.48% (Mean ± STD). To have a better understanding of error distribution, the average confusion matrix over 15 subjects was calculated (Table 2). Recognition errors were only observed when the actual mode was squat lifting. The percentages of misrecognized squat lifting as no-lift, stoop lifting and right-asymmetric lifting were 0.67%, 0.22% and 0.71%, respectively. All the stand-to-sit movements were misrecognized (transitioned from *Pre-extension* to *Extension*) in the first step of the algorithm. The accuracy of no-lift recognition indicates the percentage of misrecognized stand-to-sit movements being successfully corrected in the second step. The average sensitivities across 15 subjects for no-lift, squat lifting, stoop lifting, left-asymmetric lifting and right-asymmetric lifting were 100.00 ± 0.00%, 98.40 ± 2.25%, 100.00 ± 0.00%, 100.00 ± 0.00% and 100.00 ± 0.00%, respectively. The average specificities for no-lift, squat lifting, stoop lifting, left-asymmetric lifting and right-asymmetric lifting were 98.82 ± 0.37%, 100.00 ± 0.00%, 99.94 ± 0.23%, 100.00 ± 0.00% and 99.81 ± 0.54%, respectively.

For subject-independent recognition, the optimal feature set was $\left\{\alpha_{trunk},\delta_{LR},\delta_{thigh},\;\psi_{adj}\right\}$. The overall recognition accuracy across 15 subjects was 99.34 ± 0.85%. Though the accuracy was slightly lower (by 0.31%) than that of subject-dependent recognition, the difference was not statistically significant (*p* = 0.186). Table 3 reports the confusion matrix describing the recognition performance. Compared to the confusion matrix for subject-dependent recognition, apart from squat lifting, recognition errors were also observed when the actual mode was no-lift, stoop lifting, and right-asymmetric lifting. Misrecognition of no-lift as squat lifting had the highest error rate, which was 1.33%. For the other misrecognitions, the error rates were lower than 1%. The average sensitivities across 15 subjects for no-lift, squat lifting, stoop lifting, left-asymmetric lifting and right-asymmetric lifting were 98.67 ± 3.99%, 98.81 ± 2.20%, 99.29 ± 2.77%, 100.00 ± 0.00% and 99.76 ± 0.92%, respectively. Compared with subject-dependent recognition, the differences in sensitivity were not statistically significant for all the classes (*p* = 0.217, 0.603, 0.334, 1.000 and 0.334 for no-lift, squat lifting, stoop lifting, left-asymmetric lifting and right-asymmetric lifting, respectively). The average specificities for no-lift, squat lifting, stoop lifting, left-asymmetric lifting and right-asymmetric lifting were 99.94 ± 0.23%, 99.74 ± 0.77%, 99.81 ± 0.54%, 99.81 ± 0.74% and 99.87 ± 0.34%, respectively. Compared with subject-dependent recognition, the differences in specificity were not statistically significant for all the classes (*p* = 0.171, 0.217, 0.416, 0.334 and 0.668 for no-lift, squat lifting, stoop lifting, left-asymmetric lifting and right-asymmetric lifting, respectively).

### 3.2. Lifting Detection Delay

The average absolute detection delay over 15 subjects was 166 ± 41 ms, 134 ± 27 ms, 136 ± 29 ms, and 121 ± 20 ms for squat, stoop, left-asymmetric and right-asymmetric lifting, respectively (Figure 4A). Since lifting speed (the average hip joint angular speed from $t_{peak}$ to $t_{2}$) varied significantly across subjects (from 55.6 to 91.9 deg/s, 47.0 to 66.9 deg/s, 50.3 to 82.8 deg/s and 54.1 to 83.8 deg/s for squat, stoop, left-asymmetric and right-asymmetric lifting, respectively), the absolute detection delay data were normalized to investigate the impact of lifting techniques on lifting detection delay.

The average normalized detection delay over 15 subjects was 14.2 ± 2.6%, 11.1 ± 1.8%, 11.4 ± 2.0%, and 10.5 ± 1.5% of total lifting time for squat, stoop, left-asymmetric and right-asymmetric lifting, respectively (Figure 4B). According to the result of Shapiro-Wilk’s test, the data of normalized detection delay was normally distributed, as *p* > 0.05 for all the lifting techniques. The result of one-way repeated measures ANOVA indicated that the normalized detection delay was significantly different between lifting techniques (*p* < 0.001). Pair-wise comparisons between squat lifting and the other three lifting techniques were all significant (*p* = 0.002 for squat vs. stoop, *p* = 0.002 for squat vs. left-asymmetric, and *p* < 0.001 for squat vs. right-asymmetric). The pair-wise comparisons between stoop, left-asymmetric, and right-asymmetric lifting were not statistically significant (*p* = 1 for stoop vs. left-asymmetric, *p* = 0.564 for stoop vs. right-asymmetric, and *p* = 0.311 for left-asymmetric vs. right-asymmetric).

## 4. Discussion

Understanding the user’s lifting technique is important for active exoskeletons to provide assistance efficiently when needed, as kinematics and dynamics of human movement vary significantly for different lifting techniques. However, few studies have presented lifting detection algorithms for application with wearable exoskeletons, and most of them rely on surface EMG signals collected from hip flexor-extensor muscles or forearm muscles to control the assistive action of the robot [40,41]. In a recent study by Cevzar et al., a finite state machine with Gaussian mixture models (GMM) was developed to classify five kinds of movements (i.e., forward bending, squatting, walking, stair climbing, and chair sitting) using signals of an IMU and two encoders, for the purpose of controlling a quasi-active hip exoskeleton [42]. Whenever forward bending was classified, the passive actuator mechanism was engaged to assist the user movement, whereas the actuators where disengaged when performing the other movements. However, classification results for this study were not reported. Moreover, to the best of the authors’ knowledge, the present study is the first to report the use of lifting technique recognition to provide ad-hoc pre-defined assistive profiles.

The primary contributions of this study are the exploration and confirmation of the promise of real-time lift technique recognition for the purpose of exoskeleton control. The proposed algorithm has two main features, which may present advantages over previous approaches: first, it relies on sensory information collected by exoskeleton-embedded sensors, without relying on extra wearable sensors or biosignals; second, it can recognize the lifting technique performed by the user.

Based on the algorithm performance results of this study, subject-dependent and subject-independent technique recognition accuracy (both > 99.3%) and onset recognition delay (<170 ms) proved appropriate for use in application with wearable robots. Among the investigated lifting techniques, the algorithm had significantly higher normalized detection delay in squat lifting than in other lifting techniques. This finding may be attributed to the additional time taken by subjects to adjust their posture prior to lifting, since it is usually more difficult to maintain balance for squat lifting [30]. Computation time of the algorithm is also included in the absolute detection delay, but it is very small. The average computation time of the algorithm (including both step 1 and step 2) over 1172 detections is 0.29 ± 0.05 ms in MATLAB platform. Though in the present study the proposed algorithm was not tested for online control of the exoskeleton, according to the results achieved in our previous study [25], this amount of detection delay would not introduce noticeable discomfort to users.

While in our previous study [25] we also asked subjects to perform freestyle lifting with no specified technique, we did not consider freestyle lifting in this study because there is no explicit definition for freestyle lifting, and we found that subjects typically adopted a hybrid technique combining elements of squat and stoop lifting. The four lifting techniques tested in this experiment showed obvious differences in signal characteristics, which were helpful for lifting technique recognition. However, sensor signals of each lifting technique also exhibited some between-subject variance, which was probably due to the differences in subjects’ anthropometry, ways of performing different lifting techniques, and of wearing the device. The subject-related variance challenged the algorithm to perform subject-independent recognition because features used in the recognition should not only discriminate different lifting techniques, but should also have small variance across different subjects. Therefore, a feature with large between-subject variance should be discarded, even if it is very accurate for some individual subject(s). For example, $\delta_{LR}$, $\delta_{thigh}$ and $\;\psi_{adj}$ are used in the optimal feature set of both subject-dependent and subject-independent recognitions, which means they not only can characterize different lifting techniques, but also show high similarities among different subjects. However, $\alpha_{thigh}$ is only selected for subject-dependent recognition, but discarded for subject-independent recognition. This is probably related to the use of the same height of the load for all the subjects, which would be expected to cause considerable variation in the thigh angle (when bending down to grasp the load) among subjects, according to their heights. Instead, trunk angle seems to be less influenced by subject’s height, and it can explain why $\alpha_{trunk}$ was selected in the optimal feature set for subject-independent recognition.

Despite the very good performance of the presented algorithm, the current study has some limitations, primary among which is the lack of online validation of the algorithm. Indeed, the exoskeleton was controlled in transparent mode to collect data for offline analysis, and no assistance was delivered to subjects. Future works will focus on developing assistive strategies for different lifting techniques and evaluating the performance of the algorithm for online exoskeleton control. In addition, the robustness of the proposed algorithm will be tested in scenarios more similar to real working environments.

## 5. Conclusions

In this study, we developed an algorithm for lifting onset detection and technique classification using hip exoskeleton-embedded sensors. The ability to recognize different lifting techniques enables active exoskeletons to select appropriate control strategies and provide more efficient assistance to users when performing lifting tasks. The proposed algorithm achieved reliable lifting detection performance with high recognition accuracy and relatively small detection delay. More importantly, the algorithm generalized well across different subjects, and the recognition accuracy of subject-independent recognition was 99.34%. The experimental results suggest that the developed strategy is promising for control of active exoskeletons for load lifting.

## Figures and Tables

**Figure 1 sensors-19-00963-f001:**
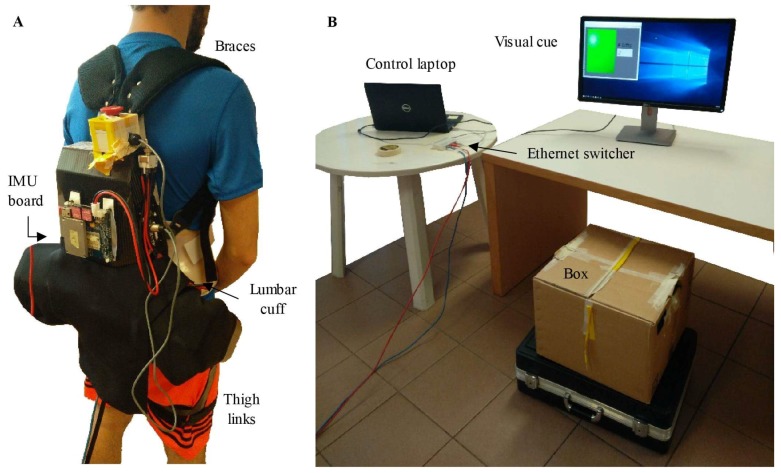
(**A**) A subject wearing the APO with an IMU board on the backpack. (**B**) Experimental setup of this study.

**Figure 2 sensors-19-00963-f002:**
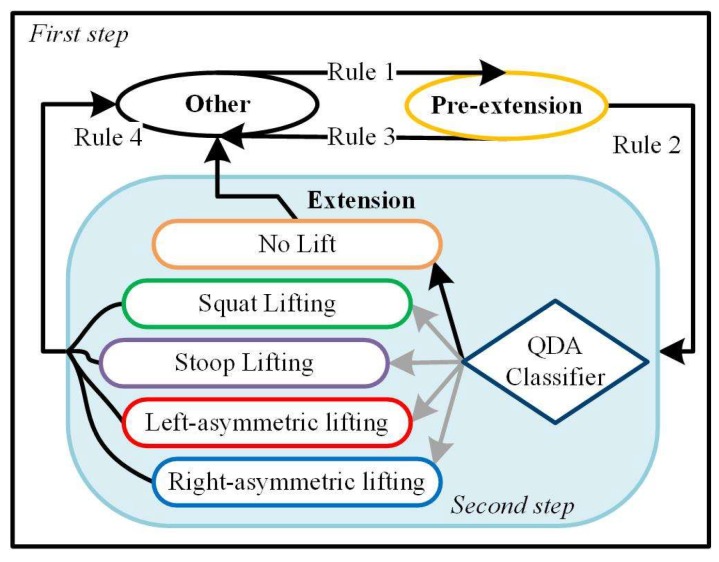
Block diagram of the recognition algorithm. The white region and the light blue region denote the first and the second steps of the algorithm, respectively. Rules 1–4 detect different phase transitions. Rule 1 refers to Equations (1)–(3); Rule 2 refers to Equations (4)–(6); Rule 3 refers to Equation (7), and Rule 4 refers to Equations (8) and (9).

**Figure 3 sensors-19-00963-f003:**
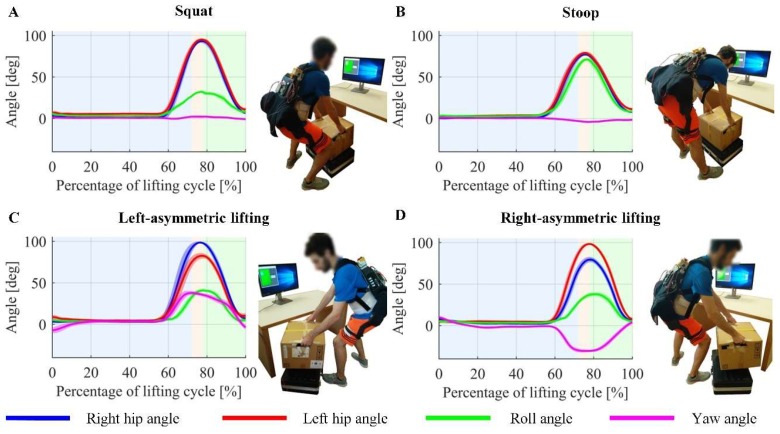
(**A**–**D**) Sensor signals of squat, stoop, left-asymmetric and right-asymmetric lifting over the lifting cycle for a representative subject. The figures were plotted using the data for lifting the load and placing it on the table, which were collected in Session 1 of the experiment. Solid curves and shaded regions denote mean values and standard deviations of corresponding angles over multiple cycles, respectively. The three classification outputs are denoted by backgrounds of different colors: *Other* (blue), *Pre-extension* (yellow) and *Extension* (green). Left and right hip angles are measured by exoskeleton’s hip joint encoders, and roll and yaw angles are estimated using raw signals collected by the IMU on exoskeleton’s backpack.

**Figure 4 sensors-19-00963-f004:**
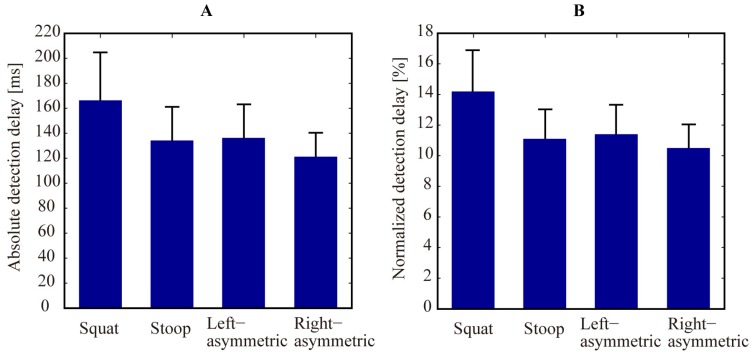
Lifting detection delay of the algorithm. (**A**,**B**) show absolute and normalized detection delay for four different lifting techniques, respectively. Error bars denote STDs of the delay across 15 subjects.

**Table 1 sensors-19-00963-t001:** List of candidate features.

Feature	Definition and Description
$\alpha_{hip}$	$\theta_{mean}\left(i\right)$, left and right average hip flexion angle
$\alpha_{trunk}$	φ(i), trunk flexion angle
$\alpha_{thigh}$	$\theta_{thigh}\left(i\right)$, thigh flexion angle
$\delta_{LR}$	$\theta_{L}\left(i\right)-\theta_{R}\left(i\right)$, difference between left and right hip flexion angle
$\sigma_{thigh}$	Standard deviation of $\theta_{thigh}$ from $t_{0}$ to i
$\delta_{thigh}$	$\theta_{thigh}\left(i\right)-\theta_{thigh}\left(t_{0}\right)$, difference of $\theta_{thigh}$ from $t_{0}$ to i
$\;\psi_{adj}$	$\psi \left(i\right)-\bar{\psi_{0}}\left(i\right)$, adjusted yaw angle of the IMU

Note: $t_{0}$ and i denote the initial moment of the *Pre-extension* phase and *Extension* phase; φ and ψ denote roll and yaw value estimated with raw IMU signals; $\theta_{thigh}=\theta_{mean}-\varphi$; $\bar{\psi_{0}}$ is the mean value of ψ in the 500-ms analysis window before $t_{0}$.

**Table 2 sensors-19-00963-t002:** Confusion matrix (Mean ± STD) for subject-dependent recognition (%).

		Estimated Mode				
		No-Lift	Squat	Stoop	Left-Asymmetric	Right-Asymmetric
**Actual mode**	**No-lift**	100.00 ± 0.00	0.00 ± 0.00	0.00 ± 0.00	0.00 ± 0.00	0.00 ± 0.00
	**Squat**	0.67 ± 1.38	98.40 ± 2.25	0.22 ± 0.86	0.00 ± 0.00	0.71 ± 2.02
	**Stoop**	0.00 ± 0.00	0.00 ± 0.00	100.00 ± 0.00	0.00 ± 0.00	0.00 ± 0.00
	**Left-asymmetric**	0.00 ± 0.00	0.00 ± 0.00	0.00 ± 0.00	100.00 ± 0.00	0.00 ± 0.00
	**Right-asymmetric**	0.00 ± 0.00	0.00 ± 0.00	0.00 ± 0.00	0.00 ± 0.00	100.00 ± 0.00

**Table 3 sensors-19-00963-t003:** Confusion matrix (Mean ± STD) for subject-independent recognition (%).

		Estimated Mode				
		No Lift	Squat	Stoop	Left-Asymmetric	Right-Asymmetric
**Actual mode**	**No lift**	98.67 ± 3.99	1.33 ± 3.99	0.00 ± 0.00	0.00 ± 0.00	0.00 ± 0.00
	**Squat**	0.24 ± 0.92	98.81 ± 2.20	0.48 ± 1.26	0.00 ± 0.00	0.48 ± 1.26
	**Stoop**	0.00 ± 0.00	0.00 ± 0.00	99.29 ± 2.77	0.71 ± 2.77	0.00 ± 0.00
	**Left-asymmetric**	0.00 ± 0.00	0.00 ± 0.00	0.00 ± 0.00	100.00 ± 0.00	0.00 ± 0.00
	**Right-asymmetric**	0.00 ± 0.00	0.00 ± 0.00	0.24 ± 0.92	0.00 ± 0.00	99.76 ± 0.92

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
