# Peer review of "Classification of Lifting Techniques for Application of A Robotic Hip Exoskeleton"

_sensors, 2019, doi:10.3390/s19040963_

Round 1
Reviewer 1 Report
This paper presents a real-time classification algorithm of lifting techniques to be applied to hip exoskeletons. The presented results show a high level of accuracy at a reasonable computing time and delay. The analysis has been done off-line, but future works will test the proposed algorithm on-line for a better validation of the classification method.
The paper is well written and structured. The proposed method is clearly described, the experimental setup is very detailed and the results are deeply analyzed.
I have a few minor comments:
1 - This work extends previous work in which the authors were able to detect the lifting movement. In the present work, a second phase is added in order to classify the lifting technique, and also to discard wrong detections from the first phase. However, it is not clear why two phases are needed. Two consecutive phases do not have a negative effect in the delay?
2 - The differentiation of the lifting technique can be useful for the selection of the control strategy. While it is clear that the differentiation between lifting and no-lifting is a critical issue, it is not clear the effect of the application of a different control strategy for different lifting techniques. While this analysis is out of the scope of this paper, it could be interesting to give some hints.
3 - The estimation of the thigh angle is not clearly described. Is it the different between the trunk angle from the IMU and the hip angle from the respective encoder? Please, clarify this point in the manuscript.
4 - In the statistical analysis of the results, the mean and SEM are used. However, it would be more significant to use the standard deviation instead of the SEM. This will give a better understanding of the dispersion of the data.
Author Response
This paper presents a real-time classification algorithm of lifting techniques to be applied to hip exoskeletons. The presented results show a high level of accuracy at a reasonable computing time and delay. The analysis has been done off-line, but future works will test the proposed algorithm on-line for a better validation of the classification method.
The paper is well written and structured. The proposed method is clearly described, the experimental setup is very detailed and the results are deeply analyzed.
Thank you very much for the kindly comments.
I have a few minor comments:
1 - This work extends previous work in which the authors were able to detect the lifting movement. In the present work, a second phase is added in order to classify the lifting technique, and also to discard wrong detections from the first phase. However, it is not clear why two phases are needed. Two consecutive phases do not have a negative effect in the delay?
Thank you. As we mentioned in the manuscript, in the first step of the algorithm, it is easy to make wrong detections during stand-to-sit movement. Therefore, it is necessary to have a second step to discard wrong detections. Actually, in our previous work, the algorithm also has two steps. The main improvement in the current work is that we use a new approach in the second step, which enables the algorithm to not only discard wrong detections but also classify the lifting technique. Of course, extra delay will be introduced due to the second-step algorithm. However, the introduced delay does not cause noticeable discomfort to users, while a high level of detection accuracy can be guaranteed, which is very important for safety considerations.
2 - The differentiation of the lifting technique can be useful for the selection of the control strategy. While it is clear that the differentiation between lifting and no-lifting is a critical issue, it is not clear the effect of the application of a different control strategy for different lifting techniques. While this analysis is out of the scope of this paper, it could be interesting to give some hints.
Thank you for the comment. I think the effect of applying the control strategy of a lifting technique for a different technique depends on how different these techniques are in kinematics and dynamics. If the difference is relatively small, maybe it will just introduce a little discomfort. But if the difference is large enough, it might cause dangers to the user.
3 - The estimation of the thigh angle is not clearly described. Is it the different between the trunk angle from the IMU and the hip angle from the respective encoder? Please, clarify this point in the manuscript.
Thanks a lot. Hip joint angle is defined as (180° - the angle between the trunk and the thigh). The thigh angle is the angle of thigh with respect to the vertically downward direction. Similarly, the trunk angle is the angle of trunk with respect to the vertically upward direction. We provide more details in the revised manuscript. Please find it in line 179 – 183.
4 - In the statistical analysis of the results, the mean and SEM are used. However, it would be more significant to use the standard deviation instead of the SEM. This will give a better understanding of the dispersion of the data.
Thank you for the suggestion. In the revised manuscript, standard deviations are reported.
Reviewer 2 Report
The paper describes the development of a classifier of potential interest for controlling exoskeletons in case of lifting operations. The authors consider 4 different lifting techniques; they collect data from healthy subjects while wearing an exoskeleton and lifting 5 kg. Data are only obtained from available joint sensors and IMU.
The classification starts using rules, in practice a set of dis-equations with empirical thresholds. In the first step it considers only the hip joint angle to discriminate pre-extension, or other movements, and uses a set of 4 rules to detect the transitions. In the second step, as soon as the first step outputs extension, statistical method (QDA) with 7 features (angles or simple equations on angles) is used to recognize 5 classes (4 lifting or no lift).
The classifier is analyzed in terms of accuracy both for each subject and using external test data. The detection delay is also analyzed, and the reported times are compatible with a controller frequency. The results show that the classifier is worth further investigation to be really use by the exoskeleton controller.
There are a few problems that require attention:
- Why using only one load, and why 5 kg? Please explain the reasons.
- How many are the initial features available from the sensors? Please provide a sentence or a table.
- How many are the features really used by the classifier in the second step? In Table 2 there is the indication that 4 features were optimal, while 7 features have been obtained using feature reduction. Please clarify.
- The final statistics presented are about the full classifier (step1 and 2)? Why only accuracy and detection delay are reported? Sensitivity and specificity could be of interest too.
- There is never an indication about the computational times of the method. In figure 4 the times in ms of the delay are reported; please indicate how much is given to the acquisition and how much to the computation, and whether the computation has been done considering the exoskeleton controller or not.
Author Response
The paper describes the development of a classifier of potential interest for controlling exoskeletons in case of lifting operations. The authors consider 4 different lifting techniques; they collect data from healthy subjects while wearing an exoskeleton and lifting 5 kg. Data are only obtained from available joint sensors and IMU.
The classification starts using rules, in practice a set of dis-equations with empirical thresholds. In the first step it considers only the hip joint angle to discriminate pre-extension, or other movements, and uses a set of 4 rules to detect the transitions. In the second step, as soon as the first step outputs extension, statistical method (QDA) with 7 features (angles or simple equations on angles) is used to recognize 5 classes (4 lifting or no lift).
The classifier is analyzed in terms of accuracy both for each subject and using external test data. The detection delay is also analyzed, and the reported times are compatible with a controller frequency. The results show that the classifier is worth further investigation to be really use by the exoskeleton controller.
Thank you very much for the comment.
There are a few problems that require attention:
- Why using only one load, and why 5 kg? Please explain the reasons.
Thank you. We used only 5-kg load for two reasons. First of all, this study is mainly focused on validating the proposed algorithm. We do not want to use too heavy load that might cause injuries to subjects. Second, in our pilot study, we asked subjects to perform lifting tasks with both 5-kg and 10-kg loads. However, we did not find significant differences in lifting kinematics and lifting detection performance between experiment trials using these two loads.
- How many are the initial features available from the sensors? Please provide a sentence or a table.
Thank you. With the two hip joint encoders and one IMU, we mainly used four channels of signals (namely left hip angle, right hip angle, and roll and pitch angle of the trunk IMU) for feature extraction. By visually inspecting the signals, 7 features were calculated (see Table 2) to generate the initial feature pool. Though these details have already been included in the manuscript, maybe they are not clear enough. To avoid misunderstanding, the details have been better clarified in the revised manuscript (line 175 – 179).
- How many are the features really used by the classifier in the second step? In Table 2 there is the indication that 4 features were optimal, while 7 features have been obtained using feature reduction. Please clarify.
Thank you. As we mentioned above, we calculated 7 features (see Table 2) to generate the initial feature pool. Then a greedy-forward selection approach was used reduce the number of features (from 7 to 4) for classification, and the finally determined feature set for classification is named as optimal feature set. Therefore, 4 features were actually used by the classifier in the second step. This issue was better clarified in the revised manuscript (line 200 – 201).
- The final statistics presented are about the full classifier (step1 and 2)? Why only accuracy and detection delay are reported? Sensitivity and specificity could be of interest too.
Thanks a lot for the comment. Yes, the final statistics refer to results obtained with both step 1 and step 2 of the algorithm. According to the reviewer’s suggestion, we reported the sensitivity and specificity for each class as well in the revised manuscript (line 230 – 233, line 278 – 283, and line 293 – 303).
- There is never an indication about the computational times of the method. In figure 4 the times in ms of the delay are reported; please indicate how much is given to the acquisition and how much to the computation, and whether the computation has been done considering the exoskeleton controller or not.
Thank you. The delay reported in Figure 4 is the delay respect to the timing of hip angle peak. Computation time is also included in the delay, but it is quite small. The average computation time of the algorithm (including both step 1 and step 2) over 1172 detections is 0.29 ± 0.05 ms in MATLAB platform. The exoskeleton controller runs at 100 Hz, which means signal acquisition, lift technique recognition and all the other data processing in the control loop together take less than 10 ms. The result of computation time is included in the revised manuscript (line 359 – 361).
Reviewer 3 Report
This paper shows the validation of an algorithm for classifying different lifting tasks based on embedded sensors in a hip active exoskeleton.
From a scientific point of view, the work is interesting and the topic dealt with is quite relevant nowadays, having, potentially, great applicative impact in healthy people but also in aging and disabled people.
The study was conducted with great methodological rigor and the results shown are robust and the conclusions are consistent.
I just suggest to the authors to modify the title so to better let understand that the validation, at this moment, was offline. The term "real-time" could made confusion to readers.
In conclusion, in the opinion of the revisor there are no particular lacks to address, except for those already highlighted by the authors for the future, concerning the online validation of the classifier.
Author Response
This paper shows the validation of an algorithm for classifying different lifting tasks based on embedded sensors in a hip active exoskeleton.
From a scientific point of view, the work is interesting and the topic dealt with is quite relevant nowadays, having, potentially, great applicative impact in healthy people but also in aging and disabled people.
The study was conducted with great methodological rigor and the results shown are robust and the conclusions are consistent.
I just suggest to the authors to modify the title so to better let understand that the validation, at this moment, was offline. The term "real-time" could made confusion to readers.
In conclusion, in the opinion of the revisor there are no particular lacks to address, except for those already highlighted by the authors for the future, concerning the online validation of the classifier.
Thank you very much for the kindly comments. According to your suggestion, the term “real-time” has been removed from the title in the revised manuscript.